# Hydroclimate Trend Analysis of Upper Awash Basin, Ethiopia

Fekadu Aduna Duguma [1,2,*], Fekadu Fufa Feyessa [1], Tamene Adugna Demissie [1] and Krystyna Januszkiewicz [3]

1    Faculty of Civil and Environmental Engineering, Jimma Institute of Technology, Jimma University,
     Jimma P.O. Box 378, Ethiopia; fekadu.fufa@ju.ed.et (F.F.F.); tamene.adugna@ju.edu.et (T.A.D.)
2    Department of Civil Engineering, College of Engineering and Technology, Nekemte Campus,
     Wollega University, Nekemte P.O Box 395, Ethiopia
3    Faculty of Civil Engineering and Architecture, West Pomeranian University of Technology in Szczecin,
     50 Piastów Ave., 70-311 Szczecin, Poland; krystyna_januszkiewicz@wp.pl
*    Correspondence: fkdaduna861@gmail.com

**Abstract:** The Awash River basin is classified into the upper basin, middle basin, and lower basin. The upper basin is the most irrigated and socio-economically important, wherein early and modern agriculture started. This study aimed to assess the upper basin's hydroclimate variability under climate change from 1991 to 2015 following the county's land-use policy change. Distinguished topographical settings, namely, lowland, midland, and highland, are used for upper Awash basin hydroclimate trend analysis. Lowland stations revealed a nonsignificant seasonal and annual increasing trend except for the Autumn season. Midland stations showed a decreased seasonal rainfall. Except for Sendafa, the increased station, the highland area exhibited an annual decreasing trend. The Awash-Hombole and Mojo main tributaries are used for the evaluation of basin streamflow. The Awash-Hombole main tributary resulted in annually growing trends during the summer season. Mojo main tributary resulted in a significantly decreasing trend during the spring, summer, and autumn seasons with a 99% level of significance. Therefore, following the basin's topographic nature, the change of hydroclimatic elements, mainly of the rainfall and streamflow, is observed. Accordingly, its hydroclimate variated by 11 and 38% with precipitation and streamflow, respectively, from the mean value within the study time series.

**Keywords:** hydroclimate; upper awash basin; trend analysis; streamflow

## 1. Introduction

African Rainfall is characterized by significant variability in precipitation [1] leading to supply challenges for drinking and irrigation water. Worldwide, societies benefit from a better understanding of natural and anthropogenic changes in rainfall [2]. The Hydroclimate of East Africa shows distinctive variabilities on seasonal to decadal timescale, meaning that changes from large-scale climate variability, including the El Niño Southern Oscillation (ENSO), Indian Ocean Dipole (IOD), and movement of the inter-tropical convergence zone (ITCZ) [3–6] directly influence the hydroclimate of Ethiopia and Awash basin specifically.

The country is considered to have abundant water resources potential, and further, some see it is an East African water town. However, it is facing floods and drought from poor water resource management with reluctant policy. Hence, essential water resource development with optimal utilization is vital for sustainable agriculture-related economic development [7]. The country has 12 basins, and the main rivers of these basins are transboundary waters, except for the Awash River and the Rift lakes [1,8–10]. Claim global climate change has put the water resources of basins under hydro-climatic variability, resulting in a significant socio-economic challenge. Nowadays, the Awash basin is known for high climate variability involving droughts and floods; climate change will intensify the existing challenges [11]. The Ministry of Water Resources and Energy introduced the Awash Valley Authority (AVA) in 1962 to solve persistent problems of basin water stress, develop an infrastructural plan, and develop the basin water resources

administration [12,13] AVA failed to provide its full mandate from institutional turnover, altering its socio-political situation [12].

The year 1991 saw a political shift in the country in which the socialist government was overthrown by the Ethiopian People's Revolutionary Democratic Front (EPRDF), who declared the country to be the Federal Democratic Republic of Ethiopia (FDRE). Following this change, the newer constitution announced by December 1994 proclaimed that 'Land is a common property of the nations, nationalities and peoples of Ethiopia and shall not be subject to sale or other means of transfer' [14]. Illegal land-use change and urban expansion [10,14,15], which will be covered in the following article, have put stress on the basin's water resource.

Historically, the first hydroelectric dam named Aba Samuel was built in 1932 in the Upper Awash basin, and later in 1960 it was named Koka Reservoir. Awash River basin is a basin wherein modern agriculture was introduced in 1950s. This basin is the most irrigated basin in the country, such that it is the economically most important basin of the country [16]. The basin is water-stressed from intensive irrigations and domestic water supply, mainly with its tributes from April to June [12,17]. Likely, the river basin's discharge has decreased since 1981 [18]. Climate change and water resource fluctuation are inseparable natural events. Whether it is quantifiable or not, the difference in these events might result in hydroclimate change, which could devastate the socio-economy. Based on the land use dynamism from urban expansion within the upper basin, the most progressive climate change resulted in water stress [11] and agricultural economic tension.

There is high confidence that developing countries will be more vulnerable to climate change than developed countries [19,20]. Accordingly, different studies have shown that the hydroclimate of the Awash basin changed based on the rainfall simulating satellites data [5,11,21–23].

Therefore, assessment and quantification the hydroclimate change is essential in water resource planning and management. It is the fundamental motivation to investigate the relationships between climate and water resources [8,24] Accordingly, this study aimed to assess the Upper Awash basin's hydroclimate change under existing climate stress concerning its topographic classes.

## 2. Methods

### 2.1. Description of the Study Area

Scholars classify the Awash basin into three parts: upper basin, middle basin, and lower basin, according to its climatological, physical, socio-economic, agricultural, and water resource characteristics [11,25]. Upper Awash is a river basin that crosses the Ethiopian plateau to the central rift, and it covers 11,402 km$^2$. This river flows from above 3000 to 1600 m a.m.s.l from its headwaters at Ginchi 75 km west of Addis Ababa to the Koka Lake. This basin is delaminated with the Abay River basin from the western side, the Omo-Gibe, Rift Valley Lakes Basin to the southwest, and the Wabe Shebele River basin to the southeast [26]. The basin extends from 38.52 to 53.23 longitude and 8.79 to 10.26 latitude, according to Figure 1.

The Upper Awash basin is part of the Ethiopian climate mainly controlled by the seasonal migration of the Intertropical Convergence Zone (ITCZ) and associated atmospheric circulations related to its complex topography [27]. The country has a diverse climate ranging from semi-arid desert types in the lowlands to humid and warm (temperate) classes in the southwest. Accordingly, the Awash basin ranges from moist in its upper basin to arid in its lower basin. This basin has a mean annual temperature of 27.18 °C and mean 25.87 and 28.98 °C minimum and maximum temperature [18].

Upper Awash river crosses some parts of the Ethiopian plateau via Bacho plain to the Koka Reservoir in the main Ethiopian rift floor. Addis Ababa, the capital, and other major towns are located in the basin.

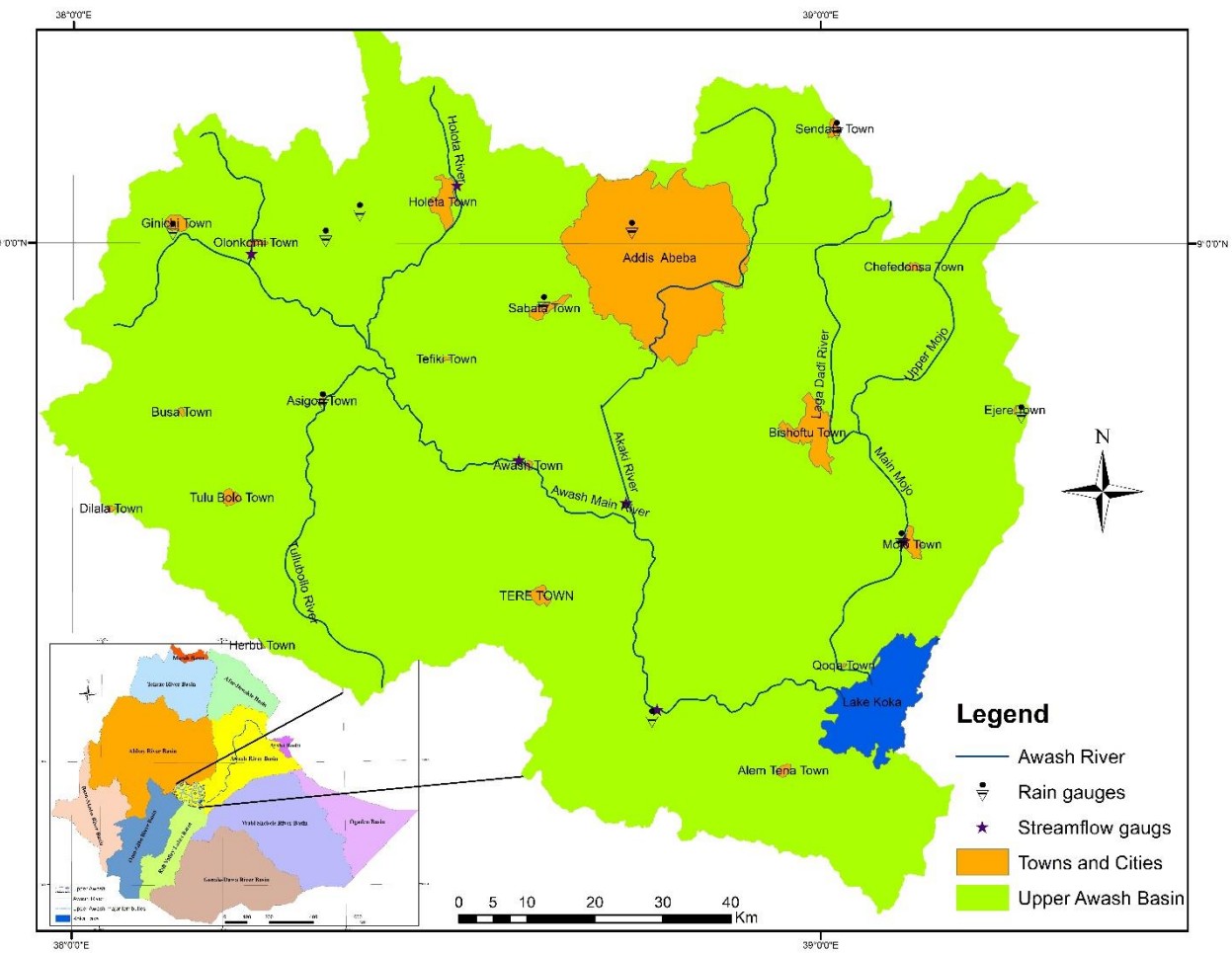

**Figure 1.** Upper Awash basin location map.

### 2.2. Data and Analysis Method

The basin daily hydroclimate data elements of 25 years (1991 to 2015) study time series were subjected to statistical evaluation. These data elements were collected from 11 meteorological stations and seven streamflow stations. These were obtained from National Meteorological Agency (NMA) (see Table 1) and the Ministry of Water, Energy, and Irrigation (MWEI) (see Table 2); they were considered for trend analysis according to data consistency.

The stations' daily data inconsistency occurred commonly in the Awash basin. Stations had acceptable monthly missed data; the trend evaluation is performed according to Section 2.2.1. Accordingly, stations with the mean monthly missing value greater than 15% were rejected from the analysis.

Traditionally, the climate of the Awash basin has four seasons: Arfaasaa refers to Spring (March, April, and May), Ganna is Summer (June, July, and August), Birraa is Autumn (September, October, and November), and Bona is Winter (December, January, and February). Arfaasaa and Ganna are wet seasons, while Birraa and Bonaa are dry seasons, relatively speaking. The heavy rainy season extends to September of the Birraa season.

The primary trend analysis starts from the central concept that climate change is possibly manifested in the difference in precipitation, resulting in streamflow regime change. Thus, the evaluation of the Upper Awash basin's streamflow fluctuations are analyzed from the correlation between the basin's precipitation and streamflow during the study time series. The basin streamflow trend evaluation was performed within Awash-Hombole and Mojo main tributaries as the Koka Reservoir inlets are mojo and Awash-Hombole river streams.

**Table 1.** Precipitation stations with their location monthly missing values.

| SN. | Station | Longitude | Latitude | Elevation (M) | Missing | Missing % |
|-----|---------|-----------|----------|---------------|---------|-----------|
| 1 | Addis Ababa | 38.75 | 9.02 | 2386 | 1 | 0.24 |
| 2 | Addis Alem | 38.38 | 9.042 | 2372 | 6 | 1.41 |
| 3 | Asgori | 38.33 | 8.79 | 2072 | 15 | 3.53 |
| 4 | Ejere | 39.27 | 8.77 | 2254 | 28 | 6.59 |
| 5 | Bishoftu | 38.95 | 8.73 | 1900 | 10 | 2.35 |
| 6 | Ginci | 38.13 | 9.02 | 2132 | 5 | 1.18 |
| 7 | Hombole | 38.77 | 8.37 | 1743 | 60 | 14.12 |
| 8 | Kimoye | 38.34 | 9.01 | 2150 | 3 | 0.71 |
| 9 | Mojo | 39.11 | 8.61 | 1763 | 10 | 2.35 |
| 10 | Sabata | 38.63 | 8.92 | 2220 | 19 | 4.47 |
| 11 | Sendafa | 39.02 | 9.15 | 2569 | 51 | 12.00 |

**Table 2.** Streamflow gauging stations.

| SN. | Station name | Longitude | Latitude | Area (km$^2$) | Period |
|-----|--------------|-----------|----------|---------------|--------|
| 1 | Akaki | 38.47 | 8.53 | 884.4 | 1991–2009 |
| 2 | Awash Ballo | 38.41 | 8.85 | 2568.8 | 1991–2012 |
| 3 | Berga | 38.21 | 9.10 | 248 | 1991–2012 |
| 4 | Holota | 38.51 | 9.08 | 316 | 1991–2011 |
| 5 | Melka Kunture | 38.36 | 8.42 | 4456 | 1991–2015 |
| 6 | Mojo | 39.50 | 8.36 | 1264.4 | 1991–2015 |
| 7 | Hombole | 38.47 | 8.23 | 7656 | 1991–2015 |
| 3 | Berga | 38.21 | 9.10 | 248 | 1991–2012 |
| 4 | Holota | 38.51 | 9.08 | 316 | 1991–2011 |
| 5 | Melka Kunture | 38.36 | 8.42 | 4456 | 1991–2015 |
| 6 | Mojo | 39.50 | 8.36 | 1264.4 | 1991–2015 |
| 7 | Hombole | 38.47 | 8.23 | 7656 | 1991–2015 |

2.2.1. Filling Missed Data

Stations' monthly data discrepancy is observed from its daily collection, and the filling method is carried out. The missed data filling was based on a percent of missing monthly data during the study time series. Some studies [28–31] employed various methods to fill missed hydrometeorological data to perform the statistical analysis.

Stations with monthly missing data not more than 15% were subjected to normal numerical ratio and simple linear regressions [28,30]. The linear regression assumes that values are missing at random, and it is an essential technique that forms the basis of many elaborate algorithms [32,33]. Hence, stations with <15% and >10% missing gaps are filled with the application of the numerical normal ratio method, and stations having missing value <10% were filled with simple linear regression from the nearby stations [30] according to Equations (1) and (2), respectively.

$$Rs = \frac{\overline{R}s}{n} + \sum_{i=1}^{n} \left( \frac{Ri}{\overline{\overline{R}i}} \right) \tag{1}$$

where $\overline{R}s$ and $\overline{R}i$ are annual average rainfall at a station with missing value and neighboring gauges, respectively; $Ri$ is the rainfall in the adjacent meter; and $n$ is the number of neighboring stations.

$$Px = \frac{1}{n} + \sum_{i=1}^{n} \frac{Pi}{n} \qquad (2)$$

where $Px$ is a missing monthly value, $Pi$ is the neighbor's record, and $n$ is the neighboring stations' number.

2.2.2. Homogeneity and Correlation Test

The homogeneity test is another process used to understand the consistency data of stations. It was tested with Pettitt's test, the Standard Normal Homogeneity Test (SNHT), Buishand's method, and Von Neumann's test. The cumulative deviation was used to detect the inhomogeneity [34]. The possibility of a homogeneity test from the partial sum or cumulative deviation from mean is noted by [35], according to the following Equation (3).

$$Sk^* = \sum_{i=1}^{k}(yi - \overline{y}), \; k = 1, 2, \ldots \ldots n, \qquad (3)$$

The term $Sk^*$ is the partial sum of the given series. If there is no significant change in the mean, the difference between $yi$ and $\overline{y}$ will fluctuate around zero ($y$ is the annual series and $\overline{y}$ is the mean). Ref. [34] calculated significance of the change in the mean value with 'rescaled adjusted range' $R$, which is the difference between the maximum and the minimum of the $Sk^*$ scaled values by the sample standard deviation:

$$R = \frac{\max Sk^* - \min Sk^*}{SD}, \; 0 \le k \le n \qquad (4)$$

Autocorrelation from repeated data determines the randomness and influences of the hydroclimatic time series trend. Calculation of change using modified Mann Kendall trend analysis is when the lag-1 autocorrelation exists. Different studies discuss the occurrence of autocorrelation, which is used to reduce the possible exaggeration in the significance of trend analysis, or it may lead to the underestimating of the trend. For example, [36] and [37] discussed Error I (error of accepting the null hypothesis) and Error II (error of rejecting the null hypothesis). Therefore, the pre-whitening method is the most common method to solve problems related to autocorrelation. Pre-whitening significantly reduces the false detection of a trend when it does not exist [36]. The autocorrelated hydroclimate data ware was removed from serial dependence prior to Mann Kendall trend evaluation with the pre-whitening application.

*2.3. Mann Kendall Statistical Model*

Nonparametric Mann Kendall/modified Mann Kendall [38] and Sen's Slope [39] are commonly applied to quantify the basin's hydroclimate trend. The Mann Kendall test is a nonparametric test in which the rank of data values within a time series are compared. Mann Kendall's statistics test (Equation (5)) assumes that observations are independent and random. The test statistic $S$ and critical test statistics $Z$ are described as:

$$S = \sum_{i=1}^{n-1} \sum_{j=i+1}^{n} (Xj - Xi) \qquad (5)$$

The trend test is applied to $Xi$ data values ($i$ = 1, 2, . . . , $n$) and X ($j$ = $i$ + 1, 2, . . . , $n$). The data values $Xi$ are used as reference points to compare with the data values of $Xij$, which are given as (6):

$$sgn(S) = \begin{cases} 1, & if\ (Xj - Xi) > 0 \\ 0, & if\ (Xj - Xi) = 0 \\ -1, & if\ (Xj - Xi) < 0 \end{cases} \tag{6}$$

where $X_i$ and $X_j$ are the values in periods $i$ and $j$ when the number of data series is greater than or equal to 10 (n > 10); the MK test is then characterized by a normal distribution with the mean E($S$) = 0, and the variance $Var(S)$ is equated as (7):

$$Var(S) = \frac{n(n-1) \times (2n+5) - \sum_{k=1}^{m} tk \times (tk-1) \times (2tk+5)}{18} \tag{7}$$

where $m$ is the number of the tied groups in the time series and $tk$ is the number of data points in the $k$th tied group.

The critical test statistic or significance test $Z$ is as follows (8):

$$Z = \begin{cases} \frac{S-1}{\sigma}, & if\ S > 0 \\ 0, & if\ S = 0 \\ \frac{S+1}{\sigma}, & if\ S < 0 \end{cases} \tag{8}$$

If $Z$ > 0, it indicates an increasing trend. When $Z$ < 0, it represents a decreasing trend and a significance level at the $Z$-score's critical values where it is greater than ±1.65, ±1.96, and ±2.58 at 0.1, 0.05, and 0.01, respectively.

The original Mann Kendall test neglects the effects of serial correlation. Ref. [38] considered the autocorrelation effect and developed the Modified Mann Kendall test.

Nonparametric Mann Kendall trend analysis criteria, homogeneity, and autocorrelation analysis were considered. There were no homogeneity problems, while few data were autocorrelated. The autocorrelated data were observed within the hydroclimate data. Ref. [38] recommended the modified Mann Kendall trend analysis. The Akaki and Mojo stations' annual and Bona streamflow of Awash Bello, Berga, and Hombole stations' data exhibited autocorrelation. Therefore, these data were subjected to modified Mann Kendall trend analysis (See Section 4.1).

### 2.3.1. Sen's Slope

Nonparametric Sen slope quantifies existing increasing or decreasing statistical trends. Furthermore, the Sen slope estimator is used to evaluate the amount of time series trend change [14] when it is assumed to be linear [34], according to Equation (9).

$$D(t) = Q(t) + B \tag{9}$$

where $D(t)$ is an increasing or decreasing trend function of time, $S$ is a slope, and $B$ is intercepted (constant).

Thus, the slope of each data pair $Q_i$ is expressed according to Equation (10):

$$Q_i = \frac{X_i - X_j}{j - i} \tag{10}$$

where, $j > i$, i.e., $X_j$ is a data reading time series $j$, while $X_i$ is the data reading time series.

### 2.3.2. Topography and Hydroclimate Change

Basin change of rainfall and temperature stations evaluation was within the elevation ranges from about 3400 to 1577 m a.m.s.l. Topographic classification of the study area revealed from DEM (Figure 2) with a spatial resolution of 12.5 m from Earth Data Alaska

Land Facility https://earthdata.nasa.gov/ (accessed on 3 March 2019) using ArcGIS 10.3. This topographic variation has three systematical classifications, according to Figure 2. The first class is lowland, with elevation ranges from 1577 to 1990 m a.m.s.l; the midland class is elevated from 1990 to 2250 m a.m.s.l, and the highland area is the catchment elevated from 2250 to 3400 m a.m.s.l. The basin's topographic formation is described, and accordingly, the basin's temporal data analyses for the hydroclimate trend evaluation are carried out (Section 4).

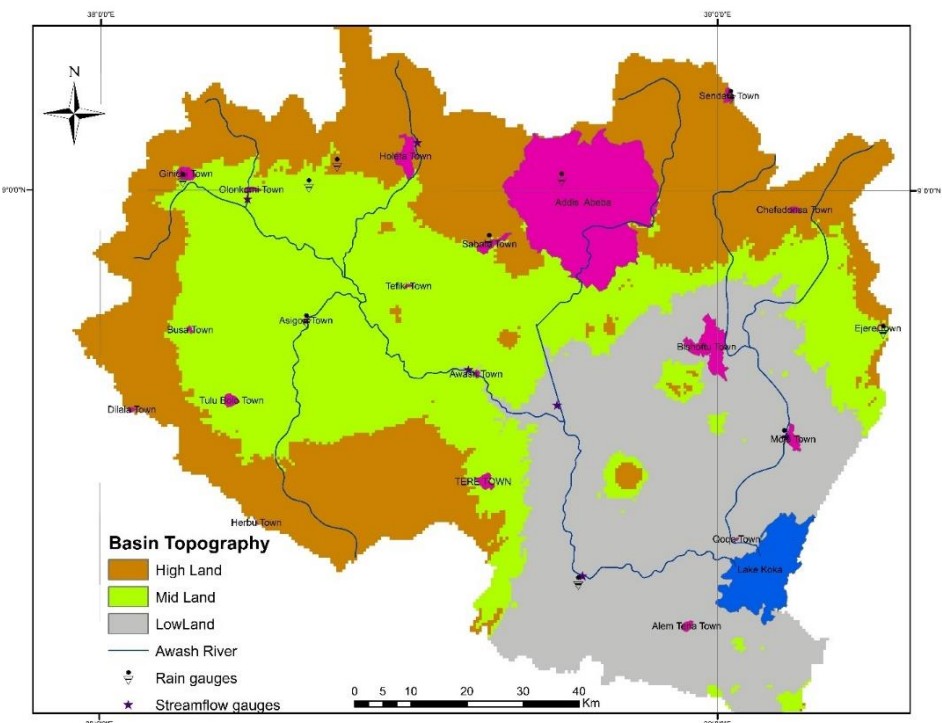

**Figure 2.** Elevation based classified map of upper Awash Basin.

### 2.3.3. Regression Analysis

The importance of this analysis is to detect trends in basin hydroclimate concerning its topographic classes correlations that relapsed with the multiple linear regression method. The model developed from maximum temperature, minimum temperature, mean streamflow, and mean precipitation in the basin. According to Equation (11), the simple linear regression calculation is considered topography to precipitation correlation from the meteorological station's elevation value to its mean annual precipitation variability element.

$$D = S_i + R_i \overline{R_b} \tag{11}$$

$D$ is basin mean streamflow, $S_i$ stream flow intercept, $\overline{R_b}$ basin mean rainfall, and $R_i$ rainfall intercept; the basin rainfall condition will control change in the streamflow.

## 3. Result and Discussion

The upper Awash basin had an elevation of approximately 1800 m difference on average from the basin river head to its outlet. The elevation variation was calculated from the satellite image digital elevation model (DEM). The basin's hydroclimate analysis with multiple linear regression, elevation to temperature, and precipitation to elevation correlation was performed. The systematic elevation classification had a different mean of rainfall. Accordingly, the lowland had about 361 mm, the midland got 354 mm, and the highland had about 137 mm mean annual rainfall during the study time.

The statistical application was used to correlate the basin's topographical nature to its temporal data. Accordingly, the linear regression of topographic classes to hydroclimate

correlation agreed with the finding of [39] (See (Section 4.2. Hence, the temperature variability revealed from trend analysis association to precipitation evaluation carried out.

## 4. Meteorological Trend Analysis

The precipitation data from eleven selected stations for the 1991 to 2015 time series (Figure 3) were used to evaluate the meteorological trend. The basin had 1019 mm mean annual rainfall. It had 1340 mm maximum and 873 mm minimum rainfall by 1993 and 1997, respectively. Arfaasaa, Ganna, and Birraa use agricultural time, and these are seasons considered to encompass basin's most essential time. Therefore, seasonal and annual hydroclimate trend evaluation with respect to basin topography nature must be emphasized.

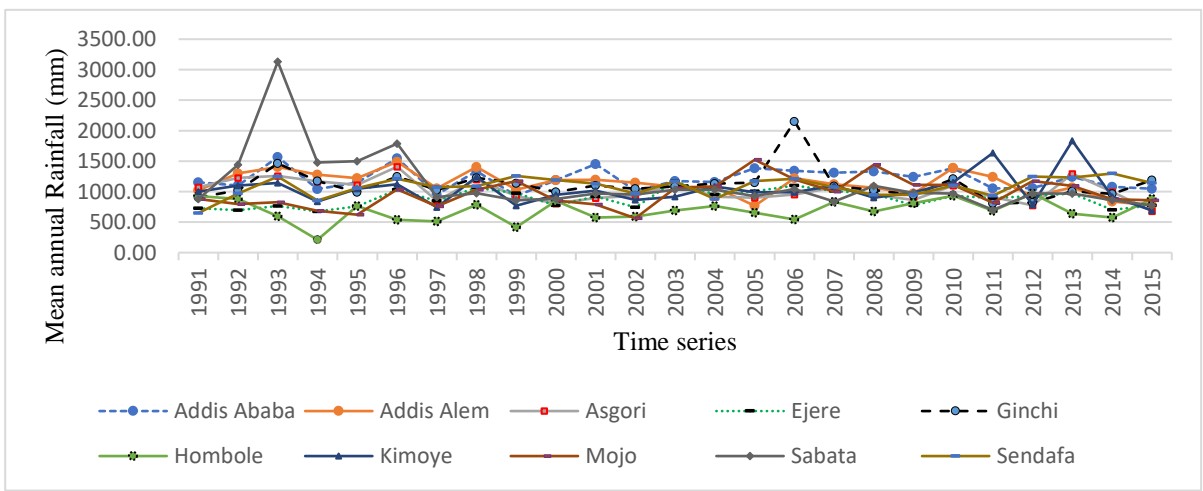

**Figure 3.** Upper Awash basin mean annual rainfall from 1991 to 2015.

The lowland (Bishoftu, Mojo, and Hombole) stations displayed an increasing hydroclimate trend, both seasonally and annually (Figure 4c). They had an annual significant increasing trend with a 95% statistical significance.

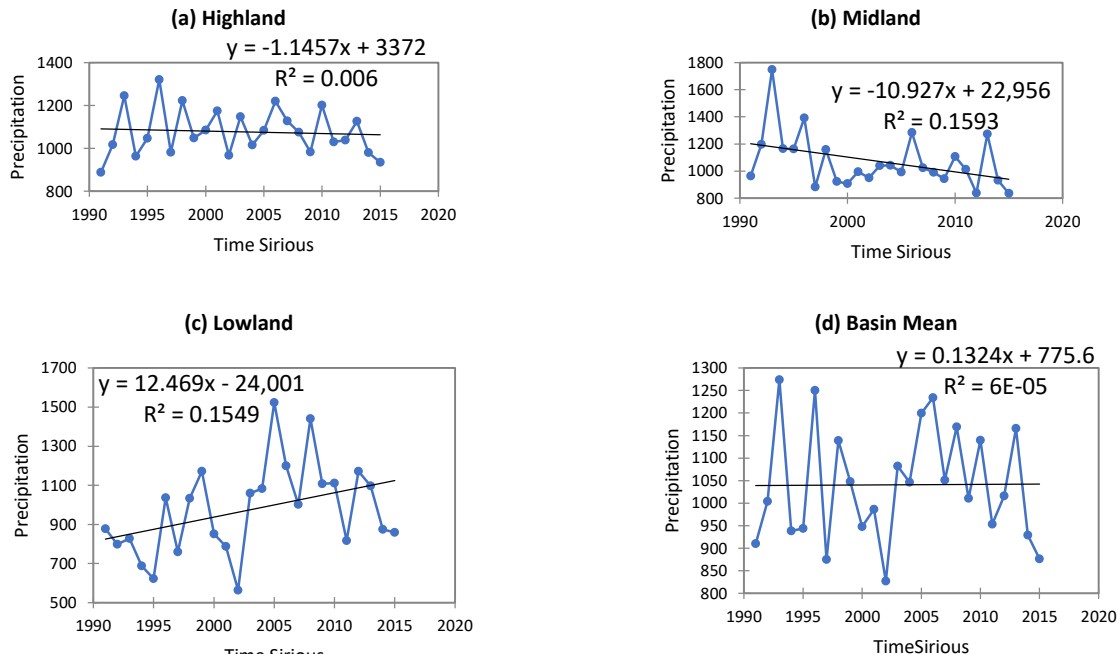

**Figure 4.** Topographic based precipitation trend (Y: shows the slope of the trend and $R^2$: tells the degree of correlation); (Figure **a–d** are the Topographic based rainfall within the respective study time serious).

The Hombole and Mojo stations that contribute 31.22% (Table 3) of basin precipitation had insignificant seasonal rainfall growth except for the Birraa season of Mojo. The Bishoftu station had an insignificant hydroclimate rise seasonally and annually during the study time series. The Birraa season of Mojo experienced a rainfall and temperature increase at 95 and 90% significance levels, respectively.

**Table 3.** Topographical rainfall contribution.

| Variable | Minimum | Maximum | Mean | Std. Deviation | Rainfall Contribution % |
|---|---|---|---|---|---|
| Highland | 888.19 | 1321.13 | 1077.06 | 109.11 | 34.50 |
| Midland | 835.10 | 1748.20 | 1070.35 | 201.47 | 34.28 |
| Lowland | 564.30 | 1523.40 | 974.80 | 233.18 | 31.22 |
| Basin Mean | 827.06 | 1273.87 | 1040.74 | 126.77 | 100.00 |

Midland (Sabata, Kimoye, Ginchi, and Asgori) stations that accounted 34.28% (Table 3) precipitation of the basin had decreasing seasonal precipitation (Figure 4b). Annually, Sabata station significantly increased to a 90% level of significance with rainfall.

The maximum temperature at Kimoye and Asgori stations increased, while Sabata and Ginchi stations decreased contrarily. However, Ginchi and Sabata stations insignificantly decreased with maximum temperature.

Precipitation of highland (Addis Ababa, Sendafa, Ejere, and Addis Alem) area stations exhibited a decreasing trend annually and in Arfaasaa seasons. These area stations contributed 34.5% of basin's precipitation (Table 3). During Ganna season, Addis Alem and Addis Ababa stations increased insignificantly. Annually, these stations increased at 95% significance level. However, Ejere stations decreased considerably at 95% significance level. Generally, highland stations exhibited an annually decreasing trend except for Sendafa station.

Since highland stations lack temperature data, the Addis Ababa station is considered for the trend analysis. Thus, it resulted in an increasing trend annually and in all seasons with least correlation of $R^2$ (Figure 4a). Statistically, the annual trend of maximum temperature and seasons' (Ganna and Birraa) minimum temperature significantly increased. The detailed analysis is expressed in (Table 4).

**Table 4.** Trend analysis result of precipitation.

| Station | Season | S-Value | p-Value | Z-Value | Sen's Slope | CV (%) |
|---|---|---|---|---|---|---|
| Addis Ababa | Arfaasaa | −34 | 0.392 | −0.864 | −0.781 | 38.99 |
| | Ganna | 36 | 0.418 | 0.817 | 0.851 | 12.73 |
| | Birraa | 24 | 0.59 | −4.825 | 0.518 | 34.33 |
| | Bona | −37 | 0.4 | −3.315 | −0.223 | 81.07 |
| | Annual | −14 | 0.764 | −0.303 | −0.201 | 14.57 |
| Addis Alem | Arfaasaa | −24 | 0.595 | −0.537 | −0.315 | 33.36 |
| | Ganna | 32 | 0.442 | 0.888 | 0.959 | 55.63 |
| | Birra | −14 | 0.76 | −0.303 | −0.302 | 58.27 |
| | Bona | −141 | 0.001 | −0.223 | −3.539 | 144.62 |
| | Annual | −98 | 0.023 | −2.26 | −1.028 | 17.03 |
| Bishoftu | Arfaasaa | 50 | 0.25 | 1.14 | 0.96 | 54.9 |
| | Ganna | 52 | 0.23 | 1.19 | 1.15 | 20.56 |
| | Birra | 44 | 0.32 | 1.004 | 0.4 | 29.03 |
| | Bona | −26 | 0.56 | −0.585 | −0.07 | 125.51 |
| | Annual | 66 | 0.13 | 1.52 | 7.27 | 18.1 |

**Table 4.** *Cont.*

| Station | Season | S-Value | p-Value | Z-Value | Sen's Slope | CV (%) |
|---|---|---|---|---|---|---|
| Ejere | Arfaasaa | −36 | 0.414 | −0.817 | −0.784 | 47.25 |
| | Ganna | −132 | 2.17 | −3.06 | −3.267 | 18.27 |
| | Birra | 11 | 0.8 | 2.336 | 0.08 | 14.97 |
| | Bona | −90 | 0.04 | −2.088 | −0.63 | 14 |
| | Annual | −84 | 0.052 | −1.938 | −0.975 | 15.78 |
| Ginchi | Arfaasaa | −34 | 0.441 | −0.771 | −0.825 | 46.75 |
| | Ganna | −16 | 0.726 | −0.35 | −0.634 | 25.76 |
| | Birra | −34 | 0.0446 | −0.771 | −0.43 | 32.28 |
| | Bona | −72 | 0.097 | −1.67 | −0.47 | 93.25 |
| | Annual | −28 | 0.528 | −0.631 | −0.311 | 22.98 |
| Hombole | Arfaasaa | 0 | 0.981 | 0 | 4.861 | 78.39 |
| | Ganna | −68 | 0.117 | −1.564 | −2.027 | 39.8 |
| | Birra | 60 | 0.721 | 1.38 | 0.72 | 62.38 |
| | Bona | 21 | 0.64 | 4.672 | 0.007 | 259.85 |
| | Annual | −6 | 0.907 | −0.117 | −0.018 | 39.64 |
| Kimoye | Arfaasaa | −10 | 0.833 | −0.21 | −0.296 | 41.85 |
| | Ganna | −18 | 0.691 | −0.397 | −0.18 | 11.18 |
| | Birra | −22 | 0.628 | −0.49 | −0.29 | 82.73 |
| | Bona | −72 | 0.71 | −0.374 | −0.196 | 193.48 |
| | Annual | −10 | 0.833 | −0.21 | −0.047 | 24.64 |
| Mojo | Arfaasaa | 46 | 0.293 | 1.051 | 0.89 | 62.84 |
| | Ganna | 36 | 0.414 | 0.817 | 1.676 | 35.34 |
| | Birra | 82 | 0.058 | 1.892 | 0.78 | 48.37 |
| | Bona | −4 | 0.943 | −0.072 | 0.001 | 113.11 |
| | Annual | 66 | 0.129 | 1.518 | 0.747 | 27.28 |
| Sabata | Arfaasaa | −29 | 0.313 | −0.654 | −0.85 | 67.87 |
| | Ganna | −4.84 | 0.009 | −2.6 | −3.041 | 45.9 |
| | Birra | −18 | 0.695 | −2.335 | −0.34 | 29.4 |
| | Bona | −83 | 0.056 | −0.374 | −0.73 | 18.22 |
| | Annual | −86 | 0.035 | −2.11 | −1.124 | 43.85 |
| Sendafa | Arfaasaa | 2 | 0.981 | 0.023 | 0.033 | 64.77 |
| | Ganna | 88 | 0.042 | 2.031 | 2.861 | 21.93 |
| | Birra | −73 | 0.088 | −1.682 | −1.104 | 51.05 |
| | Bona | −67 | 0.117 | −1.546 | −0.481 | 92.62 |
| | Annual | 52 | 0.233 | 1.191 | 0.556 | 14.45 |
| Asgori | Arfaasaa | 55 | 0.207 | 1.262 | 0.63 | 51.35 |
| | Ganna | 70 | 0.107 | 1.611 | 1.84 | 16.49 |
| | Birra | 20 | 0.661 | 4.437 | 0.248 | 31.89 |
| | Bona | −47 | 0.272 | −1.075 | −0.395 | 103.45 |
| | Annual | 48 | 0.272 | 1.097 | 0.349 | 17.55 |

The basin precipitation displayed a decreasing trend of 10.58% annually (Figure 4d). The wet seasons: Arfaasaa decreased by 30.12%, while Ganna season increased by 11.26%. The basin's temperature exhibited an increasing trend. During Bona, the dry season, it got higher variability in minimum temperature and decreased precipitation. The precipitation of this season experienced variation with 72.89% decrease and increasing maximum temperature.

### 4.1. Seasonal and Annual Streamflow Evaluation

Most of the main tributaries head from the highland of the basin's central urban lands to the Koka reservoir (Figure 2). The basin receives two rivers from headwater to the lake Koka at the upper basin (Figure 5a,b). These upstream rivers were systematically approached by considering Awash Hombole and Mojo rivers, the main tributaries. Both the upper Awash Hombole and Mojo rivers cross the explained elevation difference (Figure 2).

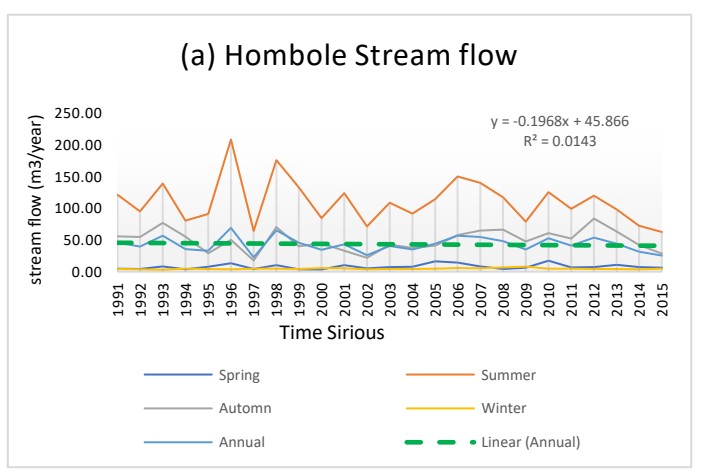 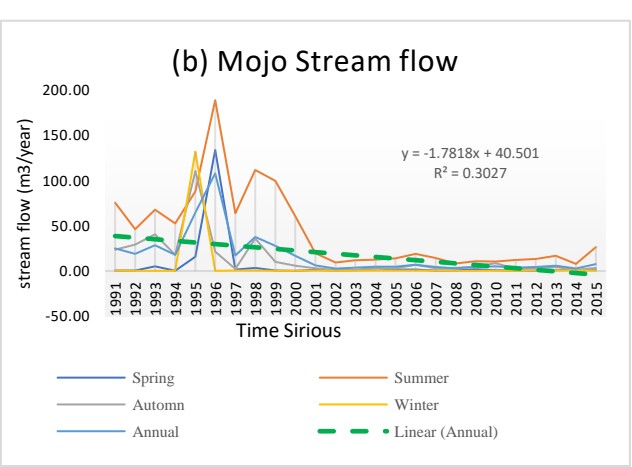

**Figure 5.** Upper Awash basin main tributaries (**a**,**b**) are the main stream flow of the Basin).

Awash-Hombole is the main tributary from Awash-Melka kunture from Awash head flow with different tributaries and joins with Akaki's main arm at Geba Robi (472,731.62 m E, 953,030.32 m N) type locality. This river crosses the three topographic classes used for the Metrologic trends. Dominantly, it flows from the highland and midland topographic study.

Berga and Akaki stations significantly correlated during the Arfaasaa season, and correlation was observed within Mojo station in Ganna season. The modified Mann Kendall trend analysis was used when significant levels of autocorrelated data were observed, and uncorrelated data analysis was with the original Mann Kendall trend test according to Table 5.

Awash-Hombole upstream tributaries exhibited a statistically decreasing trend annually (Figure 6i). Berga River significantly increased during Arfaasaa, Ganna, and Bona seasons and annually (Table 5). Contrarily, the Akaki tributary heading from the Entoto scarp, which differentiates the Abay and Awash basins [35], experienced a statistically decreasing trend seasonally and annually.

Awash Balo station experienced a significantly decreasing trend during the Arfaasaa season, while Melka kunture, Berga, Hombole, and Holota stations experienced a statistically insignificantly increasing trend.

Akaki, Hombole, and Holota stations recorded a decreasing streamflow trend, while Berga, Melka kunture, and Awash Balo got an insignificant increasing during Ganna the rainy season. During the Birraa season, Akaki station significantly decreased, and Holota station resulted in a statistically insignificant decreasing trend.

The basin resulted in an annually increasing trend within Melka Kunture, Berga, and Awash Belo. The decreasing trend is observed with Akaki, Holota, and Hombole stations. Though the basin decreased or increased with annal trend analysis, both are statistically

not significant. Generally, the tributaries of Awash-Hombole, which are Awash-Balo, Berga, and Melka-Kunture, resulted in an increasing trend annually and during the Ganna season.

From the statistical analysis of the basin during the study time, Hombole and Mojo stations decreased (Figure 6i,ii).

**Table 5.** Trend analysis result of streamflow.

| Station | Season | *S*-Value | *p*-Value | Z-Value | Sen's Slope | CV % |
|---------|--------|-----------|-----------|---------|-------------|------|
| Akaki | Arfaasaa | −25.00 | 0.363 | −0.909 | −0.097 | 105.51 |
| | Ganna | −69.00 | 0.016 | −2.379 | −2.321 | 86.33 |
| | Birraa | −95.00 | 0.006 | −3.289 | −1.053 | 99.23 |
| | Bona | −47.00 | 0.081 | −1.742 | −0.158 | 111.56 |
| | Arfaasaa | −73.00 | 0.0103 | −12.78 | −0.3821 | 99.23 |
| | Annual | −55.00 | 0.041 | −2.045 | −0.839 | 82.33 |
| Awash Ballo | Arfaasaa | −63.00 | 0.080 | −1.748 | −0.034 | 81.47 |
| | Ganna | 97.00 | 0.007 | 2.707 | 0.492 | 18.43 |
| | Birraa | 71.00 | 0.048 | 1.974 | 0.252 | 24.42 |
| | Bona | −143.00 | 0.014 | −2.446 | −0.021 | 44.60 |
| | Annual | 75.00 | 0.037 | 2.087 | 0.143 | 17.31 |
| Berga | Arfaasaa | 48.00 | 0.156 | 1.419 | 0.026 | 86.04 |
| | Ganna | 67.00 | 0.063 | 1.861 | 0.203 | 39.40 |
| | Birraa | 55.00 | 0.128 | 1.523 | 0.081 | 81.25 |
| | Bona | −20.00 | 0.959 | −0.030 | −0.004 | 194.24 |
| | Annual | 77.00 | 0.032 | 2.143 | 0.097 | 54.22 |
| Holota | Arfaasaa | 2.00 | 0.976 | 0.030 | 0.001 | 45.10 |
| | Ganna | −4.00 | 0.928 | −0.091 | −0.002 | 29.99 |
| | Birraa | −20.00 | 0.566 | −0.574 | −0.027 | 53.10 |
| | Bona | 34.00 | 0.319 | 0.996 | 0.004 | 57.67 |
| | Annual | −24.00 | 0.487 | −0.694 | −0.010 | 33.01 |
| Melka Kunture | Arfaasaa | 27.00 | 0.543 | 0.607 | 0.143 | 70.52 |
| | Ganna | 69.00 | 0.112 | 1.59 | 1.307 | 28.43 |
| | Birraa | 9.00 | 0.851 | 0.187 | 0.122 | 31.32 |
| | Bona | −40.00 | 0.333 | −0.967 | −0.0131 | 24.25 |
| | Annual | 51.00 | 0.243 | 1.168 | 0.323 | 26.86 |
| Hombole | Arfaasaa | 52.00 | 0.233 | 1.191 | 0.123 | 48.96 |
| | Ganna | −46.00 | 0.293 | −1.051 | −1.047 | 31.72 |
| | Birraa | 14.00 | 0.761 | 0.304 | 0.249 | 33.79 |
| | Bona | 38.00 | 0.359 | 0.918 | 0.024 | 21.52 |
| | Annual | −24.00 | 0.591 | −0.537 | −0.173 | 27.95 |
| Mojo | Arfaasaa | −14.00 | 0.761 | −3.036 | −0.14 | 372.31 |
| | Ganna | −56.00 | 0.172 | −1.364 | −0.483 | 104.10 |
| | Birraa | −122.00 | 0.0471 | −2.825 | −0.0839 | 165.44 |
| | Bona | 58.00 | 0.183 | 1.331 | 0.005 | 461.97 |
| | Annual | −72.00 | 0.078 | −1.761 | −0.302 | 137.47 |

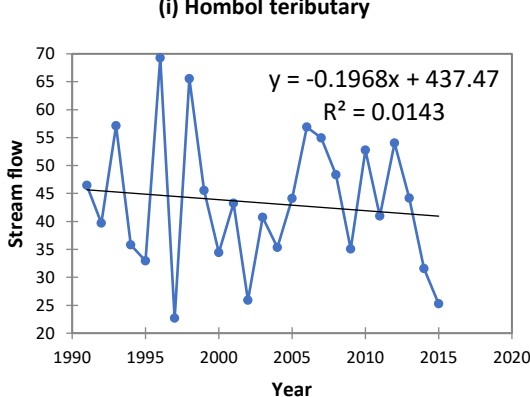

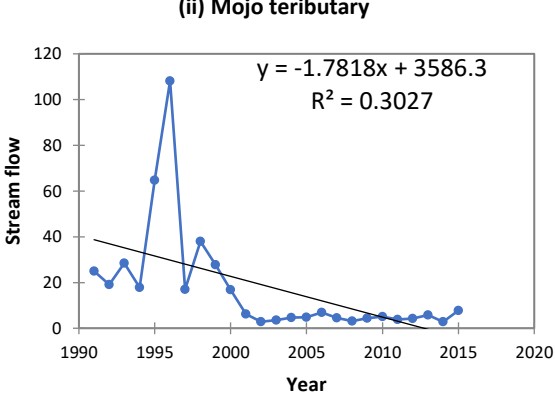

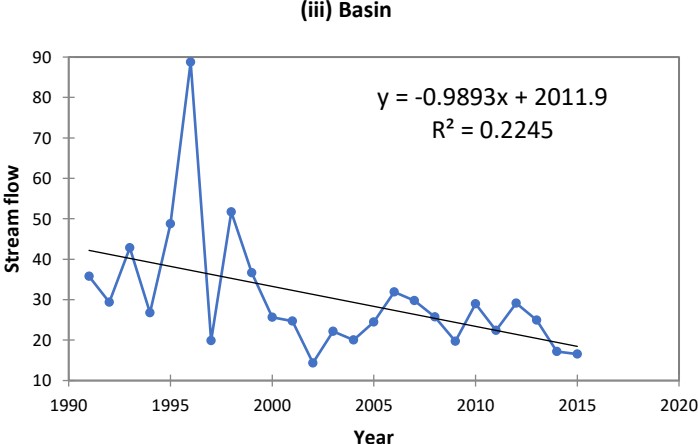

**Figure 6.** Upper Awash Basin graphical trend of stream flow; (Figure **i–iii** are the tributary based trend of the basin streamflow).

### 4.2. Rainfall and Streamflow Relationship

The linear regression method was applied to understand the correlation and rainfall influence over streamflow within the basin (Figure 7). The change in the rainfall resulted in a shift of the streamflow except for the mean annual streamflow of 1996. Accordingly, change in the rainfall and streamflow of the basin was observed, and the rainfall and streamflow of the basin varied respectively by 11.38 and 38.19% annually. The existing climate change could be one of the possible causes of such hydrologic variability of the basin.

A negative correlation between the elevation and temperature altered the basin precipitation trend. Annual mean temperature to mean annual precipitation correlated with 39.4% statistical quantification. The correlation of mean maximum temperature to mean annual precipitation and the mean minimum temperature to mean annual precipitation resulted in 48.78% and 43.37%, respectively. The statistical fitness multiple regression models were 94.47% critical coefficient and 88.94% adjusted critical coefficient.

Generally, the temperature change of Ganna and Bona of this study agreed with [11] and time trend revealed from the Global Climate Model data by [23], who got the same result studying the basin, which is a 2% decline with mean maximum temperature and streamflow during Bona, the dry season. The topographic-based temporal trend analyses revealed variability within the basin, and [21] used the same approach; hence, they got the rainfall and temperature trend variability from studying the basin.

Even though rainfall might significantly result in streamflow, the statistical correlation value between rainfall and streamflow of the Upper Awash basin fits well. The value of

$R^2$ 1 indicate that the hydroclimate element change observed in the rainfall data for the study period is strongly correlated to topographical classes of the basin.

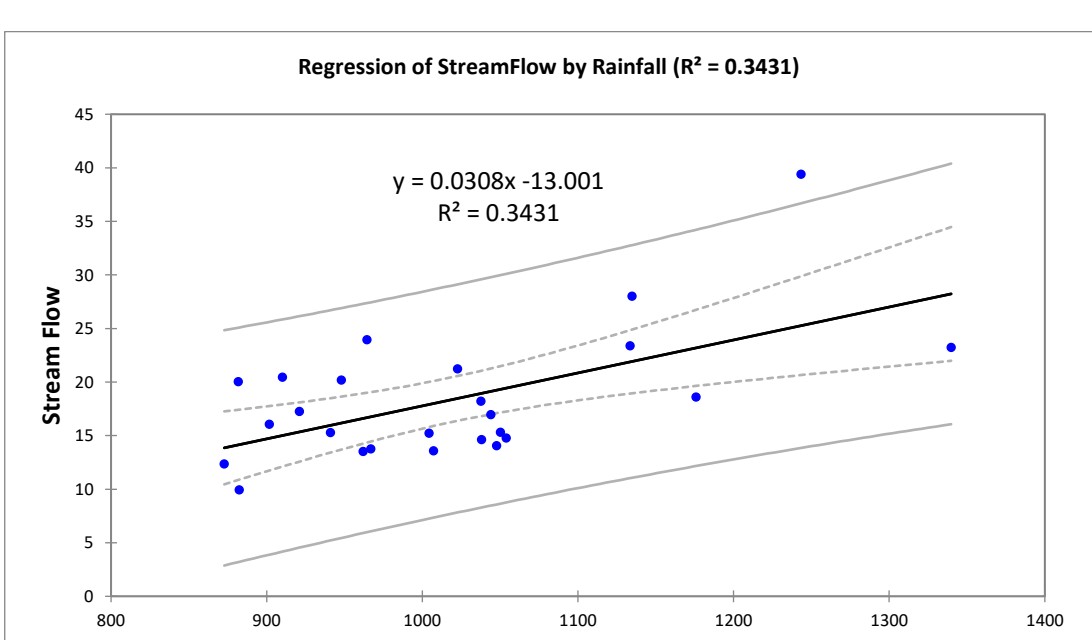

**Figure 7.** Regression of basin streamflow ($m^3$/s) by its rainfall (mm).

### 5. Conclusions

The study revealed that the basin had 1019 mm mean rainfall in the study time series. It had a mean maximum rainfall of 1340 mm in 1993 and a mean minimum rainfall of 873 mm during 1997. The station with maximum rainfall recorded was Sabata, which was 3131 mm by 1993. Hombole station had the least rainfall, 211 mm, recorded at the station in 1994. The maximum basin streamflow was 39 $m^3$/s 1996, and the minimum streamflow was 10 $m^3$/s in 2002.

The results of the basin hydroclimate trend evaluation for 25 years, from 1991 to 2015, of study period show hydrologic stress by reasonable variation. We correlated the hydroclimate elements: precipitation, temperature, and streamflow trend, with topographical classifications. The highland area got about 35%, the midland got 34.3%, and the lowland area got about 31% of the Basin's precipitation. The lowland areas showed maximum deviation in rainfall.

The historical trend correlation between topography and hydroclimate presented in Figure 5 is the first attempt at a watershed detail analysis for the Upper Awash basin. Thus, the results of the highland area meteorological evaluation show an increasing trend for precipitation and a decreasing trend for basin temperature. Gannaa season with Addis Ababa, Addis Alem, and Sandafa increased except at Sandafa station, while a decreasing trend was observed during the Arfaasaa season and annually within all stations except Sandafa. Likewise, the Addis Ababa station was considered the highland area's temperature representative; it increased from maximum and minimum temperatures. Midland stations revealed a decreasing trend in precipitation with seasonal change of climate. The lowland area resulted in an increasing precipitation trend with all stations except the Birraa season of Mojo station. Similarly, this part of the basin features a rising trend of temperature.



The Mojo river sub-basin displayed a decreasing trend of streamflow except for the Ganna season. The Awash-Hombole sub-basin displayed a decreasing trend annually except for the Berga and Melka Kunture streamflows. During Ganna season, Awash Balo, Berga, and Melka Kunture exhibited an increasing trend while the rest decreased.

Precipitation of the basin increased annually and during the Arfaasaa season in 30% of study stations (Mojo, Sendafa, and Asgori) only. The rest, 70% of them, exhibited a decreasing trend; however, Ejere and Sabata decreased insignificantly. Precipitation of the Gannaa season during study time resulted in an undulating 50% increasing and decreasing trend. Similarly, basin streamflow exhibited change. Mojo's main tributary gauging station displayed a decreasing trend in wet seasons and annually. During Arfaasaa, Ganna, and Birraa seasons, the tributary decreased, while annually and in the Birraa season, it significantly reduced at 90 and 95% significance levels.

**Author Contributions:** The listed authors made contributions. Conceptualization, methodology, and validation: F.A.D. and F.F.F. and T.A.D. Software, formula analysis, and data curation: F.A.D. and F.F.F. Admiration, Supervision, and writing—review and editing: F.A.D. and F.F.F. and T.A.D. and K.J. All authors have read and agreed to the published version of the manuscript.

**Funding:** This research received no external funding.

**Institutional Review Board Statement:** Not applicable.

**Informed Consent Statement:** Not applicable.

**Data Availability Statement:** The data that support findings of this study are available from the National Meteorology Agency. Restrictions apply to the availability of these data, which were used under license for this study. Data are available http://www.ethiomet.gov.et/ (accessed on 10 December 2017) with the permission of the National Meteorology Agency.

**Acknowledgments:** The authors thank the National Meteorological Agency, Ministry of Water and Irrigation for the preceding data provision and indebted to the Jimma Institute of Technology for funding the research.

**Conflicts of Interest:** The authors declare no conflict of interest.

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
