# Peer review of "Hydroclimate Trend Analysis of Upper Awash Basin, Ethiopia"

_water, doi:10.3390/w13121680_

Round 1

Reviewer 1 Report

Please see my attached comments

Author Response

Dear reviewer, we gladly considered your comments and we attached the answers and the corrected version of our article. Thank you.

Reviewer 2 Report

I reviewed this paper in its first version and I wrote that the work presented elements of interest that deserve publication, once it iswritten in a deeply revised version. The paper shows an analysis of hydrological data, precipitation and streamflow, collected in the upper Awash Basin basin in Ethiopia. The period examined runs from 1991 to 2015. The rainfall data of 11 stations and 7 streamflow stations were examined. Trend analyses with the Mann-Kendall method and the Sen slope were performed separately for the 4 seasons and for the whole year, accompanied by an evaluation of statistical significance. The existence of a correlation between river flows (in the closure station) and the rainfall occurring in the basin was also evaluated.

Even if some of the remarks that I pointed out have been acknowledged (revised Fig. 1 and 3, new figure 2 and a more complete Results and discussion session), I still notice that the text is written in a low level of English and presents several editorial problems and incoherence issues. Many sentences remains, in fact, incomprehensible; I strongly suggest a revision by a professional English translator.

Here I describe some remarks:

I see in my pdf copy some characters highlighted in green (lines 47, 165, 166, 167200, 203) or red (line 202); 

line 101 – 102 : Check that the centigrade degree results as 0C instead of °C;

line 166: please correct “y is the annual series” in “yi is the annual series” ;

line 180: please correct “ware” in “were”;

line 191: please correct “Xij” in “Xj”;

line 195: please correct “n_10” in “n≥10”;

line 220: in equation 9, S (slope) is missing; check if eq. 9 has to be written as D(t)=S t + B;

line 248: please correct “intercept” in “slope”;

lines 261: Peng et al. 2020, is cited but there isn’t its citation in the References chapter ;

line 273: it refers to figure 4c that is missing. In fact, the sub-plots in Figure 4 are denoted as a), b), d), e);

line 299: discussing on temperature the text recall Figure 4a that concerns precipitation;

Figure 5: the seasons are named with English name, not local names as in all the other parts of the paper.  It is preferable to use equal notations;

Line 385: add units (m3/s) to 39.4;

Finally, please check the citations and reference list. I noticed that number 23 is incomplete (I think he is Kotir 2001b).

Author Response

Dear reviewer, we thank you for your golden comments. We gladly corrected and you may find the corrected version with the attachment we did. Thank you.

Round 2

Reviewer 1 Report

The authors have now made an effort to improve the manuscript. I am ok with most of it. However, the beginning of the Introduction is still unacceptable. The precipitation trends in Africa are in fact mixed, as stated by Thomas & Nigam 2018. It's not correct to filter out one trend and staying silent on the other. Lines 27-31 really need to be changed to a more neutral tone to be consistent with current scientific understanding. I strongly recommend changing these initial two sentences to the following version:

"African rainfall is characterized by significant variability in precipitation (Thomas and Nigam 2018) leading to supply challenges for drinking and irrigation water. Societies world benefit from a better understanding of natural and anthropogenic changes in rainfall (Kotir 2011b)."

Other than that, I would be happy for the paper to be published.

Author Response

Dear reviewer, we are so thankful to your help. Please see the attached change.

This manuscript is a resubmission of an earlier submission. The following is a list of the peer review reports and author responses from that submission.

Round 1

Reviewer 1 Report

General

The text needs to be corrected by an english native speaker. It cannot be published in its current version. It also needs to be re-structured. At present, many paragraphs lack a clear logical presentation. Readers need to be better guided by the authors. The complex analyses of rainfall, river flow and temperatures need to be linguistically simpified and key results better illustrated. The proposed trends cannot be seen in Figure 3. It also becomes clear that individual datasets differ greatly from each other. So how reliable are the suggested trends actually?

One of the biggest shortfalls of the paper is that the authors do not review and use the existing literature on hydroclimatic variability in Ethiopia and East Africa. It is well known that rainfall in Ethiopia is influenced by teleconnections associated with the North Atlantic Oscillation (NAO), Indian Ocean Dipole (IOD), the Pacific Decadal Oscillation (PDO) and solar activity changes. Duguma et al. have only analyzed 15 years of data. These “modes of variability” have partly oscialltion periods of 60 years (e.g. PDO), therefore it is essential that any trends are placed in a context of natural variability.

Comments on individual parts of paper:

Abstract:

„climate change from 1991 to 2015 study time series“: Why „study time series“? Rephrase.

„The basin statistical trend analysis took place through systematic topographical classes used as Lowland, midland, and highland”

Correct to:

„The basin statistical trend analysis distinguished topographical classes, namely lowland, midland, and highland”

„except Sendafa the increased station.“: rephrase. Sentence cannot be understood.

Is it really necessary to use the Ethiopian names fro the seasons in the abstract? I suggest I stick to the english ones and mention the local name sin the main text. Also add “boreal” to seasons.

„We used to evaluate“ ???? Rephrase

Rephrase and correct this sentence:

“During study time, the change of basin streamflow within the model’s observation confidence interval of 95% except for the 1996 steam flow.”

Figure 1

Label rivrs and lakes, add some town names on main map. The names in the small inserted map are illegible. The coordinates around the map should be latitude, longitude in degree, and minutes.

Table 1

Missing values: What do you mean by that? Msising years or months or days?

Figure 2:

Scale bar should have standard units, e.g. 5, 10, 20, 30, 40 km. The coordinates around the map should be latitude, longitude in degree, and minutes.

Figure 3:

Is this annual rainfall? Ned to mention annual or season.

In text it is “Sebata” but in figure “Sabata”. Stay consistent. Please show the rainfall series separate in three stacked charts for the three topographic groups.

Page 9:

„The basin precipitation got a decreasing trend by 10.58% annually”

This is a strong statement, but I really struggle to see the justification for it. The Ganaa wet season even got wetter.

Chapter 3.2

Why are you not illustrating any time series of rive flow?

Page 11:

„Thus, the basin’s hydroclimate variated by 11.38% and 38.19% rainfall and Streamflow, respectively, from the mean value within the study time series”

What kind of information is this? It is well known that rainfall is characterized by significant natural variability. Do you mean long-term trends or just the envelope of the variation? Why would this be important? Chapter 3.3. mixes rainfall/streamflow and temperature. This is confusing.

Figure 4

More rainfall leads to more water in the river. The result makes sense, but it is rather trivial.

Concusions and Overall:

The authors describe a confusing mix of decreasing and increasing rainfall and river flow. The different trends need to be more thoroughly separated, both in text and illustrated in figures. There is nothing recognizable in Figure 3 because lines are too thick and topographic classes are not distiguished. River flow time series are not shown, why? This would be important.

Reviewer 2 Report

The paper shows an analysis of hydrological data, precipitation and streamflow, collected in the upper Awash Basin basin in Ethiopia. The period examined runs from 1991 to 2015. The rainfall data of 11 stations and 7 streamflow stations were examined. Trend analyses with the Mann-Kendall method and the Sen slope were performed separately for the 4 seasons and for the whole year, accompanied by an evaluation of statistical significance. The existence of a correlation between river flows (in an unspecified station) and the rainfall occurring in the basin was also evaluated.

Although the work has elements of interest, it is penalized by a rather low level of English and by several editorial problems. Many sentences are in fact incomprehensible. The reported results are incomplete and commented with insufficient or absent visualization. In particular, it can be noticed:  

- Figure 1 and Figure 2 show no indication of the rivers or tributaries analysed;

- The hydrometric stations are not shown in figure 2 (or even in figure 1);

- Figure 2 has an incorrect legend;

- Figure 3 does not appear significant as the tracks of the records are totally superimposed;

- Table 3 does not report the values for the Birraa season (SON) and for the whole year, which are discussed in the text, instead;

- Table 3 needs an explanatory caption of the reported values S, P, Z, Sen’s Slope, CV%;.

- Table 3 is mentioned as concerning a temperature trend analysis, but the table refers to a precipitation trend;

- Figure 4 illustrates the pairs of streamflow-precipitation values but it is not specified which fluvial and pluviometry station it is.